Multimodal system for recording individual-level behaviors in songbird groups

http://orcid.org/0000-0001-7881-3579 Rüttimann Linus 1
Wang Yuhang 1 2
http://orcid.org/0000-0002-3302-0193 Rychen Jörg 1
Tomka Tomas 1 2
Hörster Heiko 1
Hahnloser Richard H. R. 1 2 rich@ini.ethz.ch
1 Institute of Neuroinformatics, University of Zurich and ETH Zurich , Zurich , Switzerland
2 Neuroscience Center Zurich (ZNZ), University of Zurich , Zurich , Switzerland
Manjarrez Javier
Electronic publication date: 2025 Nov 10
Publication date: 2025
Volume: 13
Electronic Location ID: e20203
Received 2025 Feb 10; Accepted 2025 Sep 17
Copyright: © 2025 Rüttimann et al.
Copyright year: 2025
Copyright holder: Rüttimann et al.
License: This is an open access article distributed under the terms of the Creative Commons Attribution License, which permits unrestricted use, distribution, reproduction and adaptation in any medium and for any purpose provided that it is properly attributed. For attribution, the original author(s), title, publication source (PeerJ) and either DOI or URL of the article must be cited.
License URL: https://creativecommons.org/licenses/by/4.0/

Keywords: Computational ethology, Action recognition, Wing stroke, Multiantenna array, Phase-locked loop

Funding: National Institute of Neurological Disorders and Stroke R01NS108424 Swiss National Science Foundation 31003A_182638 and 205320_2154941 NCCR Evolving Language #51NF40_180888 Financial support was provided by the National Institute of Neurological Disorders and Stroke (R01NS108424 to Brenton G. Cooper, Richard H. R. Hahnloser, Todd F. Roberts) and the Swiss National Science Foundation: Project 31003A_182638 (the roles of vocal communication in pair formation and cultural learning in songbirds), Project 205320_2154941 (the roles of vocal communication in pair formation and cultural learning in songbirds), and the NCCR Evolving Language (Agreement #51NF40_180888). The funders had no role in study design, data collection and analysis, decision to publish, or preparation of the manuscript.

==============================
The implicit goal of longitudinal observations of animal groups is to identify individuals and to reliably detect their behaviors, including their vocalizations. Yet, to segment fast behaviors and to extract individual vocalizations from sound mixtures remain challenging problems. Promising approaches are systems that record behaviors with multiple cameras, microphones, and animal-borne wireless sensors. Good systems extract from such multimodal signals perfectly synchronized data streams. In this vein, we designed a modular system (BirdPark) for simultaneously recording small animals wearing custom low-power frequency-modulated radio transmitters. Our custom software-defined radio receiver makes use of a multi-antenna demodulation technique that eliminates data losses due to radio signal fading and that increases the signal-to-noise ratio of the received radio signals by 6.5 dB compared to demodulation from the best single-antenna. Digital acquisition of all data streams relies on a single clock, which offers accurate cross-modal redundancies that can be used to dissect rapid behaviors on time scales well below the video frame period. We demonstrate this feat by reconstructing the wing-stroke phases of free-flying songbirds and by separating the vocalizations among up to eight vocally interacting birds. In conclusion, our work paves the way for automatically dissecting complex social behaviors.

Introduction

Communication is vital for many social behaviors. However, to study interactions among animals that are kept in groups entails many measurement challenges beyond the already considerable challenges of analyzing longitudinal data from isolated animals (Kollmorgen, Hahnloser & Mante, 2020; Lipkind et al., 2017; Tchernichovski et al., 2001). One of the key difficulties of group-level behavior research is to perform automatic recognition of individuals and their actions. Actions can be recognized from video, audio, and from signals collected with animal-borne sensors. Although a single sensory modality on its own can suffice to detect certain actions, full insights into complex behavioral interactions can only be gained by combining several modalities.

Video recording stands as the quintessential and most prevalent method for monitoring many non-vocal behaviors. Video-based action recognition has traditionally been based on posture tracking (Fujimori, Ishikawa & Watanabe, 2020; Perkes et al., 2021; Segalin et al., 2020) to avoid data-hungry training of classifiers on high-dimensional video data. Recently, posture tracking has greatly improved thanks to deep-learning approaches (Badger et al., 2020; Doersch et al., 2023; Harley, Fang & Fragkiadaki, 2022; Mathis et al., 2020; Naik et al., 2023; Nath et al., 2019; Pereira et al., 2022; Walter & Couzin, 2021). Action recognition from video requires good visibility of focal animals because visual obstructions tend to hamper recognition accuracy. Given that many vocal behaviors are associated with little apparent body movement, there is a limitation to the usefulness of pure vision-based action recognition.

Sound is the modality of choice when studying acoustic communication. Sounds also signal non-vocal behaviors, for example wing flapping during flying produces a characteristic sound signature. Other example behaviors that can be recognized from sounds are preening, walking, and shaking (Stowell, Benetos & Gill, 2017). The task of classifying sounds is known as acoustic scene classification (Barchiesi et al., 2015). However, microphones record not just the focal animal but also background sounds, which makes sound-based action recognition and actor identification challenging tasks when many animals are present. Microphone arrays can improve localization and separation of sound sources (Rhinehart et al., 2020), though with arrays alone it remains difficult to assign vocalizations to individuals. Some improvements are obtained when cameras and microphone arrays are combined, to assign vocalizations to individual animals such as mice in a social home cage (Heckman et al., 2017) or dairy cattle in a barn (Meen et al., 2015). However, when animals rapidly move and simultaneously vocalize near each other, even the best microphone array coupled to a video monitoring system reach their limits in usefulness. Essentially, it remains virtually impossible to distinctly estimate the sound features of overlapping calls.

Limitations arising from sound superpositions can be overcome with animal-borne sensors such as accelerometers (Anisimov et al., 2014; Eisenring et al., 2022), gyroscopes, and microphones (Ter Maat et al., 2014). In combination with wireless transmitters (Ter Maat et al., 2014) and on-animal loggers (Anisimov et al., 2014), such sensors enable the detection of behaviors such as walking, grooming, eating, drinking, and flying, for example, in birds (Gómez Laich et al., 2009), cats (Watanabe et al., 2005), and dogs (Gerencsér et al., 2013). In general, animal-borne transmitter devices are designed to achieve high reliability, low weight, small size, and long battery life, giving rise to a complex trade-off. Among the best transmitters, in terms of battery life, size, and weight, are analog frequency-modulated (FM) radio transmitters. Their low power extends battery life and thus minimizes animal handling and associated handling stress, making them an excellent choice for longitudinal observations of small vertebrates (Gill et al., 2015, 2016; Ter Maat et al., 2014). Although sensors allow to distinctly record vocalizations of an individual, they often operate with low reliability and the vocal signal they yield is of lower quality than the signal from a stationary microphone (Stowell, Benetos & Gill, 2017), which means that to best decode animal communication, animal-borne sensor devices should be combined with other data modalities including video and sound.

Decoding animal communication from multimodal observations is still in its infancy and is afflicted with many challenges (Rutz et al., 2023). One challenge is the need to synchronize the various data streams. When each sensor modality is recorded with a dedicated device that uses its own internal sampling clock, these clocks tend to drift apart. Drifting clocks seriously hinder multimodal signal integration because changes in audio-visual latencies hamper the inference of the localization of a sound source and they prevent the detection of sound-generating motor gestures in video streams. With drifting clocks, the individual data streams must be aligned post-recording using markers in the sensor signals or auxiliary synchronization channels, which are labor intensive and error-prone processes (Dolmans et al., 2021; Pouw, Trujillo & Dixon, 2020; Zimmerman et al., 2009). To get rid of this problem, a good multimodal recording system must therefore make use of perfectly synchronized recording streams, which is the first aim of this work.

Another more particular challenge, associated with FM radio reception, is radio signal fading due to relative movements of animal-borne transmitters and stationary receivers. Fading arises when electromagnetic waves arrive over multiple paths and interfere destructively (channel fading) (Tse & Viswanath, 2005), for example by reflection off metallic walls. Fading also occurs because every receiver has a direction of zero gain, which may affect the reception of the signal from a moving transmitter. Signal fading can be addressed with antenna diversity, i.e., the use of several antennas. To combine the diverse antenna signals, typically, either the strongest signal is selected, all signals are summed, or signals are first weighted by their strength and then summed (Shatara, 2003). However, these approaches do not completely prevent fading because the signals can still annihilate. Alternatively, diversity combining is possible with phase compensation, which is the technique of rotating signal phases such that phase-compensated signals align and sum constructively (Senega, Nassar & Lindenmeier, 2017; Voitsun, Senega & Lindenmeier, 2020). Due to the limited speed of electromagnetic waves, the variable distances and orientations between a transmitter and the antennas give rise to a distinct phase offset on each antenna. Diversity combining with phase compensation reduces fading and increases the signal-to-noise ratio of the received signal, and, furthermore, it provides cues for localizing a transmitter (Berdanier & Wu, 2013; Haniz et al., 2017), which is why, as a second aim of our work, we want to bring the benefits of antenna diversity and phase compensation techniques to ethology research.

We present a custom system (BirdPark) that offers all the advantages of multimodal recordings. Birdpark is aimed at capturing behavioral gestures that either move a peripheral body part, that generate a sound, or that make the body vibrate. BirdPark features a novel multi-antenna phase compensation technique that minimizes signal losses from wireless transmitters and that improves the radio signal-to-noise ratio of single-antenna approaches. Briefly, in our diversity combining approach, we combine the signals from all four antennas such that we compensate their phase offsets to result in a mixture signal with improved signal-to-noise ratio.

We demonstrate BirdPark’s suitability for observing individual-level social behaviors in vocally interacting songbirds. The animal-borne accelerometers enable week-long monitoring of vocalizations without a change of battery. All sensor signals are perfectly synchronized, which we achieve by routing shared sampling triggers derived from a single central quartz clock to all recording devices (radio receiver, stationary microphone digitizer, and cameras). This synchronization provides the opportunity to extract from the densely sampled acoustic and vibratory signals precise timestamps of the sparse video frames, which results in a form of super-resolution imaging that we illustrate on wing flapping behaviors. We release our custom recording software and new gold standard datasets of separated vocalizations from groups of up to eight animals.

Materials and methods

Portions of this text were previously published as part of a preprint (https://www.biorxiv.org/content/10.1101/2022.09.23.509166v4.full).

Recording arena

We built an arena with two glass walls and two plexiglass walls, optimized for audiovisual recordings, minimizing acoustic resonances and visual occlusions (see Extension section, ‘Enclosures and instrumentation’) (Fig. 1A). The arena provides space for up to eight songbirds and contains nest boxes, perches, sand baths, food, and water sources (Fig. 1B). To record the sounds inside the chamber, we installed five microphones. Three video cameras capture the entire scene from three orthogonal viewpoints (the side views are recorded through the glass walls via mirrors from the ceiling). In addition, we installed a camera and microphone in each of the two nest boxes that we attached to the arena (Fig. 1B). The camera resolutions are high enough and exposure times short enough to resolve key points on birds even in mid-flight (Fig. 1D).

Figure 1 Recording arena and diagram.

(A) Inside a soundproof chamber, we built a recording arena (red dotted line) for up to eight birds. We record the animals’ behaviors with three cameras mounted through the ceiling. These provide a direct top view and indirect side and back views via two mirrors (delimited by green and purple dotted lines). To record the sounds in the chamber, we installed five microphones (blue dotted lines) among all four sides of the cage (one attached to the front door is not visible) and the ceiling, and two small microphones in the nest boxes. The radio signals from the transmitter devices are received with four radio antennas (orange dashed lines) mounted on three side walls and the ceiling. One nest box is indicated with yellow arrows and a water bottle with blue arrows. (B) A composite still image of all camera views shows two monochrome nest box views (top left) and three views of the arena (top, side, back) with eight birds among which one is flying (red arrows). Yellow and blue arrows as in A. (C) Schematic of the recording system for gapless and synchronized recording of sound (microphones), acceleration (transmitter devices), and video (cameras). The radio receiver is implemented on a USRP with a large FPGA that runs at the main clock frequency of 200 MHz. Clock dividers on the FPGA provide the sample trigger for audio recordings and the frame trigger for the cameras. The data streams are collected on a host computer that runs two custom programs, one (BirdRadio) for streaming audio and sensor signals to disk and one (BirdVideo) for encoding video data. MXI, Multisystem Extension Interface; USB3, Universal Serial Bus 3.0; PCIe, Peripheral Component Interconnect Express; TDMS, Technical Data Management Streaming; MP4, MPEG-4 Part 14; UDP, User Datagram Protocol. (D) Zoom-in on an airborne bird, illustrating the spatial and temporal resolution of the camera.

Transmitter device

All animals wear a miniature low-power transmitter device that transmits body vibrations from an attached accelerometer via an analog FM radio signal (Fig. 2A). The devices transmit the accelerometer signals as FM radio waves thanks to a coil acting as antenna (Fig. 2B). The radio circuit encodes the accelerometer signal aT(t) as (instantaneous) transmitter frequency ωT(t)≃ωc+caT(t), where t is time, ωc is the radio carrier frequency (set by the radio circuit properties), and c is some constant (see Extension section, ‘Transmitter device’). The transmitter devices weigh only 1.5 g and consume little power, resulting in a battery lifetime of about 12 days. Transmitters are firmly attached to the body using an elastic harness (Figs. 2C and 2D).

Figure 2 Transmitter device.

(A) Schematic of the electronic circuit (adapted from Ter Maat et al., 2014). The analog FM radio transmits the vibration transducer (accelerometer) signal via a high-pass filter (cutoff frequency: 15 Hz) followed by a radiating oscillator. (B) A fully assembled transmitter (left), and another one without epoxy and battery (middle), and a piece of mounting foil (right). (C) Picture of the device mounted on a bird. The transmitters are color-coded (here red) to help identify the birds in the video data. (D) Drawing of a bird wearing the device. The harness (adapted from Alarcón-Nieto et al., 2018) is made of rubber string (black) that goes around the wings and the chest.

Radio receiver

We received the FM radio signal via four antennas connected to a universal software radio peripheral (USRP) with a large field programmable gate array (FPGA), on which we implemented the eight-channel FM radio receiver. The goal of the receiver is to demodulate the transmitter signal; the receiver generates a tracking frequency ω(t) that tracks (and thereby measures) the instantaneous transmitter frequency ωT(t).

We generate the tracking frequency ω(t) using a phase-locked loop (PLL), a control system that tracks the frequency of an incoming signal. To minimize radio signal fading, we equalize the phase offsets on the four antennas with phase-compensation circuits—one circuit for each antenna. We then sum the four (complex) phase-compensated signals. The phase of the summed signal is the error signal we use to adjust the tracking frequency ω(t) of a transmitter’s PLL. This FM radio demodulation technique is described in more detail in the Extension section, ‘Multi-antenna radio signal demodulation’.

Animals and experiments

We bred and raised 20 zebra finches (Taeniopygia guttata): eight females aged 13–576 dph (day post hatch) and 12 males aged 13–1,326 dph in our colony (University of Zurich and ETH Zurich). The animals were recorded in six groups: four two-bird groups (recording duration 2–15 days), one four-birds group (recording duration 67 days), and one eight-birds group (recording duration: 89 days). No bird was used in more than one group, between recordings there were idle periods of up to several months. On each bird, we mounted a transmitter device (Fig. 2). We kept birds in the BirdPark on a 14/10 h light/dark daily cycle, with food and water provided ad libitum. We moderately tightened the harness of the sensor devices to the animals in a tradeoff between picking up vibratory signals from the singer (loose) and preserving the wearer’s natural behaviors (tight).

In this study, we were interested in evaluating the feasibility of our recording system and in benchmarking its performance, whence the small sizes of the groups with four and eight birds. The experiments were designed to study vocal communication during reproductive behaviors (groups of two birds) and during song ontogeny (groups of four and eight birds). These scientific aims will be pursued in other publications.

All experimental procedures were approved by the Cantonal Veterinary Office of the Canton of Zurich, Switzerland (license numbers ZH045/2017 and ZH054/2020). All methods were carried out in accordance with the Swiss guidelines and regulations concerning animal experiments (Swiss Animal Welfare Act and Ordinance, TSchG, TSchV, TVV). The reporting in the manuscript follows the recommendations in the ARRIVE guidelines (Kilkenny et al., 2010).

Results

Performance of multi-antenna demodulation

We validated the robustness of our multi-antenna demodulation method by analyzing the signal-to-noise ratio of the demodulated multi-antenna signal while recording a zebra finch pair over two 14 h-measurement periods. The radio signal-to-noise ratio ( RSNRM) of the complex multi-antenna signal (M) we defined as the logarithmic signal power minus the logarithm of the estimated noise power (see Extension section, ‘Radio signal-to-noise ratio’). The RSNR is a good measure of the quality of a demodulated signal because it is a monotonically decreasing function of the noise power PN of the demodulated signal estimated during non-vocal periods (Figs. 3A and 3B).

Figure 3 Multi-antenna demodulation achieves the largest RSNR.

(A) Examples of acceleration spectrograms that originate from a single calling zebra finch, with RSNRMs of 11, 22, 32, and 41 dB (top, from left to right). The bottom plots show the spectra of the accelerometer signal during calls (red lines) and noise (blue lines), computed as time averages from the spectrograms above (time segments indicated by red and blue horizontal bars). The noise power PN (integral of blue curve, units: dB re 1 Hz2), is decreasing with RSNR. When the RSNR is below 13 dB, the noise power spectral density is above the signal power of most vocalizations (these become invisible). (B) Scatter plot of PN versus RSNRM (dots), plotted for noise segments of diverse RSNRM levels. The four examples from A are highlighted (blue crossses). A polynomial of degree 5 was fitted to the data (red line). (C) Histogram of RNSR over time for the multi-antenna (RSNRM, blue line), the best single-antenna signal (RSNR*, purple line), all other single-antenna signals (grey lines), and the volumetric antenna (red line). The mean (over the measurement period of 2 × 14 h) of RSNRM (dashed blue line) is 6.5 dB larger than the mean of RSNR*(dashed purple line) and 9.0 dB larger than the mean RNSR of the volumetric antenna (dashed red line). (D) Multi-antenna demodulation is significantly better than single-antenna demodulation for the best single antenna and for the volumetric antenna, as demonstrated by the significantly longer reception periods (in percent) above a given signal-to-noise ratio. The time below the critical RSNR of 13 dB (fading signal loss) is reduced to 0% with multi-antenna demodulation.

We quantified the multi-antenna demodulation performance by comparing RSNRM to the signal-to-noise ratio RSNRa associated with each of the four single-antennas, a∈{A,B,C,D}. We found that RSNRM was about 6.5 dB higher on average than the signal-to-noise RSNR∗ of the best single antenna (Fig. 3C). The total fraction of time during which the RSNR was critically low (<13 dB, our operational definition of signal fading) using the multi-antenna demodulation was 0%, compared to 0.62% for the best single-antenna (Fig. 3D). Thus, our diversity combining approach is very effective.

To verify that simple whip antennas (a form of monopole antenna consisting of a telescopic rod) we used are no worse than more complex antennas, we recorded a test dataset of a zebra finch pair (again, two 14 h-measurement periods) with a single volumetric antenna and a single-antenna demodulation technique. The performance of the volumetric antenna was worse than that of the best simple antenna (Figs. 3C and 3D, which corroborates the effectiveness of our multi-antenna approach).

We repeated the RSNR analysis on four additional bird groups, confirming that the average RSNRM was 6.5 dB above that of the best single antenna (average across n=5 bird groups, group-average range 5.4–7.7 dB; Table 1). Similarly, we found critically low RSNRM in 0.00% (range 0.00–0.01%) of the time, compared to 2.71% (range 0.63–4.41%, n=5 pairs) of the time for the best single antenna. Thus, our technique robustly tracks frequencies even when the signal vanishes in one or several antennas.

Table 1 Signal loss statistics.

Percent time (per radio channel) during which the signal is lost due to either insufficient PLL tracking or signal fading. The latter is reported for both multi-antenna and best single-antenna demodulation. The statistics are reported for the test experiment in Fig. 3 (rows 2 and 3) and four additional experiments (copExp and juvExp).

Experiment	# Birds	Insufficient PLL tracking (%)	Fading signal: Multi antenna (%)	Fading signal: Best single antenna (%)	
TestExp (complex antenna)	2	0.41		0.01	
TestExp (multi antenna)	2	0.00	0.00	0.62	
copExpBP08	2	0.67	0.00	3.44	
copExpBP09	2	0.08	0.00	2.22	
juvExpBP01	4	0.09	0.00	4.41	
juvExpBP03	8	4.50	0.00	2.84	
Average (excl. complex ant.)		1.07	0.00	2.71	

Proximity effect and signal losses

During operation, we often observed large excursions of the tracking frequency ω(t). These excursions occurred while birds pecked on the device (Fig. 4A) or while they preened their feathers near the sensor, or when one bird sat on top of another, such as during copulations. The magnitude of these frequency excursions could reach 2.5 MHz (68 dB re 1 kHz), which is much larger than the maximal 3.4 kHz (11 dB re 1 kHz) shifts induced by vocalizations, Fig. 4B. Because these excursion occurred also without an accelerometer, Fig 4, we therefore refer to these events as proximity effects rather than reconstructed acceleration signals.

Figure 4 Large proximity effects on the transmitter frequency.

(A) On a transmitter device with a disabled (short circuited) accelerometer, the transmitter tracking frequency is strongly modulated by the proximity of the head during preening (left) and by wing movements during flight (middle). In contrast, the modulation of the (high-pass filtered) tracking frequency is much weaker for vocalizations (right) on a transmitter with enabled accelerometer. The modulation amplitudes (black arrows) due to proximity are about 1,000 times larger than the vocalization-induced modulation amplitude. (B) Distribution of modulation amplitudes for preening (n = 62), wing flaps (n = 54), and tet calls (n = 105). The preening and wing flap events were manually annotated from a set of randomly selected concatenated frequency jumps defined as data segments associated with absolute tracking frequency jumps larger than 50 kHz (see methods) and taken from a 25-h long recording of a bird with a transmitter on which we disabled the accelerometer. Tet call segments that were not masked by noise were manually annotated on a large dataset from mixed-sex zebra finch pairs.

Furthermore, although our signal demodulator tracked the transmitter frequencies very well, we observed occasional PLL tracking losses. During PLL tracking losses, the tracking frequency ω(t) associated with the given transmitter got stuck at one of the two limits of its range (see Extension section, ‘Multi-antenna radio signal demodulation’). In 14-h time periods per bird group, such tracking losses occurred at an average rate of 1.07% (range 0.00–4.50%, n=5 groups; Table 1), which was acceptably low.

Vocal segmentation analysis

Extraction of individual vocalizations from the group mixture was possible but associated with diverse challenges. On microphone channels, some vocalizations were masked by vocal signals of other birds (Fig. 5A) or by other noises (examples B1/B2 in Fig. 5B). On transmitter channels, vocal signals reached up to 7 kHz in optimal cases but only up to 1 kHz in the worst case (Fig. 5A). Furthermore, on transmitter spectrograms, some vocalizations were invisible (example A2 in Fig. 5B) and others were masked by simultaneous body movements such as wing flaps or hops (example A1 in Fig. 5B).

Figure 5 Vocal signals and their segmentation.

(A) Example spectrograms of vocalizations produced by a mixed-sex zebra finch pair. The songs and calls overlap on the sound spectrogram of a microphone channel (Mic1, top) but appear well separated on acceleration spectrograms of the male’s transmitter channel (middle) and the female’s transmitter channel (bottom). Distinct transmitter-based vocal segments are indicated by red (male) and purple (female) horizontal bars on top of the spectrograms. High-frequency vibrations appear attenuated, but even a high-pitched 7 kHz song note (white arrow) by the male is still visible. (B) Example vocalizations that are not visible in the transmitter channels (A1: syllable masked by wing flaps, A2: faint signal) or not visible on either some microphone channels (B1: faint syllable masked by noise) or all microphone channels (B2: syllable masked by loud noise). Focal TD (transmitter device) = accelerometer channel with visible vocalization (C) Crosstalk between transmitter channels can occur when birds physically touch (C1), or are near each other (C2). The red rectangles mark the regions in the spectrogram where the vocalization of the focal bird (middle row) leaks into the channel of the non-vocal bird (bottom row).

We quantified these “vocal misses” relative to a gold standard of detected vocalizations in the multimodal data stream. We segmented vocal activity by visual inspection of sound and acceleration spectrograms (Table 2, see Extension section, ‘Vocal analysis’ for details). We evaluated the number of missed vocalizations on a given channel relative to a consolidated segmentation that was based on visual inspection of all transmitter and microphone channels combined. We were particularly interested in missed vocalizations that are invisible on a given channel but visible on another channel.

Table 2 Missed vocalizations on microphone and transmitter channels.

Statistics of missed vocalizations on a transmitter channel (5th row, cf. Example A1 and A2 in Fig. 5B), on all microphone channels (6th row, cf. example B2 in Fig. 5B), and on an individual microphone channel (Mic1, 7th row, cf. example B1 in Fig. 5B). We report the number of vocalizations for which an assignment to a bird was not possible (8th row); the number of vocalizations that overlap with at least one vocalization of another bird (9th row, cf. example in Fig. 6A); the number of vocalizations with transmitter channel crosstalk (10th row, cf. examples in Fig. 5C); and the number of uncertain segments (11th row). See Methods, ‘Vocal statistics’ for details.

#	Experiment	copExpBP08	copExpBP09	juvExpBP01	juvExpBP03	Total	
2	Segmented data	1 × 7 mins	1 × 7 mins	7 × 1 min	7 × 1 min	28 min	
3	# birds	2	2	4	8	16	
4	# vocalizations	537	570	709	861	2,677	
5	# voc. missed on tr. channel	31 (5.8%)	10 (1.8%)	37 (5.2%)	18 (2.1%)	96 (3.6%)	
6	# voc. missed on all mic. channels	1 (0.2%)	3 (0.5%)	0 (0.0%)	0 (0.0%)	4 (0.2%)	
7	# voc. missed on Mic1 channel	20 (3.7%)	26 (4.6%)	1 (0.1%)	2 (0.2%)	49 (1.3%)	
8	# voc. unassigned	0 (0.0%)	0 (0.0%)	0 (0.0%)	1 (0.0%)	1 (0.0%)	
9	# voc. with overlaps	18 (3.4%)	18 (3.2%)	257 (36.2%)	353 (41%)	646 (24.1%)	
10	# voc. with crosstalk	0 (0.0%)	0 (0.0%)	30 (4.2%)	3 (0.4%)	33 (1.2%)	
11	# uncertain segments	3	3	0	0	6	

Figure 6 Wing flap imagery grouped by their latency to brief events in transmitter signals.

Following Brown’s observation of asymmetric hovering in pigeons (Brown, 1963), we categorize the wing flap cycle into eight phases, each represented by four randomly chosen examples (columns). (A) The examples’ phase bins we extracted from the time lag between the camera’s exposure window (dark red shading in A) and the dips in (event-onset subtracted) transmitter signals (n = 28 wing-flap events, see Methods, ‘Detection of radio frequency jumps’). Traces are aligned to the dip, serving as time origin for frame alignment. (B) Random video frames (n = 32) horizontally arranged in terms of their time lag, illustrating the wing flap movement.

Compared to the miss rate of 1.8% (range 0.1–4.6%, n=4 bird groups) associated with a single microphone, all microphones combined produced a reduction in miss rate by a factor of about 10, which reveals a large benefit of the multi-microphone approach. The microphone misses were mostly very short nest calls such as example B2 in Fig. 5B. The percentage of missed vocalizations on a given transmitter channel was on average 3.6% (1.8–5.8%, n=4 bird groups—note that the miss rates on microphone and accelerometer channels are not directly comparable because the former stem from two birds and the latter from a single bird). Interestingly, neither the percentage of transmitter nor microphone misses increased with the number of birds in the group. The percentage of microphone misses even decreased in larger bird groups ( n=4 and n=8 birds); however, this can be attributed to them producing fewer short nest calls, possibly due to the presence of juveniles. A very small average percentage of 0.1% (range 0.0–0.5%, n=4 bird groups) of vocalizations were missed on all microphone channels, these vocalizations were calls masked by loud noises.

The benefit of animal-borne sensing is most apparent when vocalizations overlap. We found that on average 24.1% (range 3.16–41.0%, n=4 bird groups) of vocalizations overlapped with vocalizations from another bird, which makes it virtually impossible to analyze the sound features of these vocalizations from microphone data alone. The percentage of overlapping vocalizations steeply increased with the number of birds. While microphone channels frequently picked up mixtures of vocal signals, transmitters did not in most cases (Fig. 5A). Only 1.23% of vocalizations were at most faintly visible on the transmitter of a non-focal bird. Such transmitter channel crosstalk mostly occurred when birds physically touched each other (example C1 in Fig. 5C) but sometimes also during loud vocalizations without physical contact (example C2 in Fig. 5C). In each crosstalk instance, the assignment of the vocalization to the vocalizing animal from the transmitter signal was visually straightforward.

Wing flapping behavior

Synchronized multimodal recordings enhance the precision and depth of analysis of rapid movements such as wing flapping, thanks to integration of video and accelerometer signals. Namely, the densely sampled transmitter signals, when influenced by fast behavioral events, can provide precise timing information of video frames. We demonstrate this feat by imaging the phases of the wing flap cycle on a higher temporal resolution than made available by the video frame period.

The wing flapping frequency of zebra finches ranges between 26 and 30 Hz (Ellerby & Askew, 2007). Our video frame rate of 48 Hz was insufficient to capture such nuanced motion accurately since the flapping frequency is higher than the video Nyquist frequency. Nevertheless, thanks to a randomized sampling method combining the accelerometer signal and the video frames, we were able to comprehensively image the wing flap cycle. Leaning on the transmitters’ frequency shifts caused by body movements, especially the unique shift patterns during wing flapping, we extracted from the transmitter signal the times of an images’ phase in the wing flap cycle. By sorting out the camera’s exposure window, which is considerably shorter than the video frame period, we were able to accurately assign to an image the proper wing flap phase within eight bins (Fig. 6, see Extension section, ‘Detection of wing flaps and video alignment’), i.e., at four times the temporal resolution provided by the camera’s roughly two frames per cycle. Thus, synchronized multi-modal recordings allow for the reconstruction of rapid stereotyped behaviors and they offer the potential for profound insights into fast-paced social interactions. The limiting factor of stereotyped behaviors that can be resolved is not set by the camera’s frame rate, but by the shutter speed, i.e., the available light intensity.

Discussion

We designed and validated a behavioral recording system for up to eight songbirds. We made use of analog FM radio technology that in our hands is ideally suited for continuous transmission of a single analog vibration signal. We decided to use batteries rather than power harvesting, to make our technology easily deployable in larger and more complex environments. Power harvesting is feasible in lower-power applications with low duty cycles such as optogenetics (Ausra et al., 2021; Kim et al., 2013) and it works well in rodents that navigate a small 2d environment but is more challenging in birds in a large 3D aviary since proximity between emitter and receiver is key to maximize harvested power.

We proposed a novel multi-antenna demodulation technique to minimize fading in wireless ethology research. Our phase-compensation technique increases the RNSR by 6.5 dB and reduces fading compared with single-antenna demodulation. The wireless devices transmit well-separated vocalizations unless these are either masked by large body movements such as wing flaps, they are elicited by rare cases of crosstalk, they are suppressed by poor mechanical coupling between the bird and the device, or they are lost due to insufficient PLL tracking.

We chose a relatively low radio frequency of around 300 MHz to obtain a radio wavelength of about 1 m. This wavelength roughly equals the dimension of the sound isolation chamber, implying that our radio system operates in the near field. In this regime, the relative phases of a radio wave measured on multiple antennas can be informative about the position of the transmitter and its orientation in the enclosure (Berdanier & Wu, 2013; Haniz et al., 2017), a feature that could be exploited in the future to localize the transmitters based on the relative phase information that we logged to disk. On the one hand, shorter wavelengths would make phase unwrapping techniques more prone to errors introduced by continued integration of the phase offset signal (in the context of the underlying assumption that bird movements are very smooth). On the other hand, higher frequencies might reduce the risk of resonances (standing electromagnetic waves) in the chamber and so could further improve signal-to-noise.

By combining information from transmitters and microphones, we showed that vocalization miss rates can be substantially reduced, making it possible to record the entire vocal output of an interacting group. No sensor channel by itself achieved a miss rate of less than 1%, which is a relatively large number given that we took our measurements in a relatively small, acoustically well isolated environment. We found that with increasing group size, frequent vocal overlaps prevented the comprehensive analysis of vocalizations from microphone data alone. Vocalizations appeared separated on the transmitter channels and so the more birds are recorded, the larger the benefit of animal-borne sensing.

To best of our knowledge, we are the first to report vocalization miss rates in songbird groups, as previous works have not quantified this shortcoming of group-level studies (Gill et al., 2015, 2016; Ter Maat et al., 2014). Strictly speaking, we might have missed some vocalizations even considering all the channels we used, so our estimated single-channel miss rates constitute lower bounds, implying that the true miss rate of vocalizations must be higher than what we measured. On the question about acceptable miss rates, the maximally admissible miss rate depends on the specific research question to be examined, which is why future studies will have to revisit this important measurement.

It remains open how well vocalizations can be separated from the five stationary microphones alone. In groups of mice and using microphone arrays and a camera, it has been shown that vocalizations can be assigned to individual animals by combining sound source localization and animal position tracking (Oliveira-Stahl et al., 2023). But because of frequent vocal jamming in songbirds, we doubt these approaches will make the use of animal-borne transmitters obsolete in song-learning studies to be performed in the BirdPark where subtle song changes need to be tracked in developmental time. Our dataset will hopefully be useful for the development and evaluation of such methods, as the transmitter and video channels yield well-resolved information on the individual level.

Our transmitter circuit design has several limitations to be addressed in future work. For one, we observed large fluctuations in tracking frequency during certain body movements (Fig. 4). These fluctuations were likely caused by dielectric effects near the resonator circuit. Specifically, we believe the proximity of body parts likely altered the permittivity of the space surrounding the coil, thereby modulating the carrier frequency ωc. We ascribe this problem to the single-transistor circuit that cannot be shielded because it powers the antenna that needs to radiate. This disadvantage could potentially be remedied by designing multi-transistor circuits and by shielding certain sub-circuits.

It is unlikely that the cross-talk among transmitter channels was caused by the same circuit shortcoming. Although we cannot rule out crosstalk due to inductive effects among coils caused by proximity of transmitters, the more likely explanation for the crosstalk is acoustic coupling via in-air sound waves. We therefore believe that it will not be possible to completely get rid of crosstalk. However, since in each instance of crosstalk there was a single channel standing out on which the signal amplitude was clearly maximal, the crosstalk can likely by discounted by simple operations such as intensity thresholding.

One drawback of animal-borne sensors is that they can induce negative effects on birds (Barron, Brawn & Weatherhead, 2010). In zebra finches, their effects on locomotion and singing rate are temporary: right after attachment of the sensors, the singing rate and amount of locomotion both tend to decrease but return to baseline within 3 days when sensors weigh 0.75 g (Gill et al., 2016) and within 2 weeks when sensors weigh 3 g (Anisimov et al., 2014). Given the 1.5 g weight of our devices, we expect birds to adapt to the backpacks within one week. Nevertheless, future studies will have to examine possible device-induced behaviors such as preening or pecking at the sensor.

Conclusions

We conclude that the BirdPark system is suitable for future research on the meaning of vocalizations and for research of social behaviors in general. Of key interest are courtship and reproductive behaviors that have been thoroughly studied (Caryl, 1976; Morris, 1954, 1958; Ullrich, Norton & Scharff, 2016) albeit often without examining the roles of vocal interactions during mating and pair bonding. Group-level studies in BirdPark could help us to better understand the learning strategies young birds use while they modify their immature vocalizations to match a sensory target provided by a tutor. Much of our knowledge on song learning stems from research of isolated animals in impoverished environments (Kollmorgen, Hahnloser & Mante, 2020; Lipkind et al., 2017; Tchernichovski et al., 2001), leaving many questions open about the roles of social interactions (Carouso-Peck & Goldstein, 2019; Chen, Matheson & Sakata, 2016) during this process, including by non-singing adult females (Takahashi, Liao & Ghazanfar, 2017).

To enable segmentation of animal behaviors from extensive longitudinal data, automatic methods will need to be developed. While there are many methods for single-channel segmentation available (Cohen et al., 2022; Lorenz et al., 2022; Steinfath et al., 2021) including on accelerometer data (Hoffman et al., 2024), only few combine information from multiple sensor channels (Fazzari et al., 2024; Steinfath et al., 2021) and no multimodal system includes accelerometer data so far.

Extension

Enclosures and instrumentation

The sound-isolation chamber

All our experiments were performed in a sound-isolation chamber (Industrial Acoustic Company, Winchester, UK) of dimensions 124 cm (width) × 90 cm (depth) × 130 cm (height). The isolation chamber has a special silent ventilation system for air exchange at a rate of roughly 500 l/min. Two types of light sources made of light emitting diodes (LEDs, Waveform Lighting) are mounted on aluminum plates on the ceiling of the chamber: (1) LEDs with natural frequency spectrum (consuming a total electric power of 80 W); and (2) ultraviolet LEDs of wavelength 365 nm (consuming a total electric power of 13 W).

The ceiling of the chamber contains three circular holes of 7 cm diameter through which we inserted aluminum tubes of 5 mm wall thickness that serve as holders for three cameras. The tubes also conduct the heat of the LEDs (93 W) and the cameras (13 W) to the outside of the chamber where we placed silent fans to keep the cameras below 55 °C.

Where possible, we covered the sidewalls of the chamber with sound-absorbing foam panels. A door sensor measures whether the door of the chamber is open or closed. The temperature inside the chamber was roughly 26 °C and the humidity was on average 24 ± 1% (std).

Cameras and video acquisition system

Into the three circular holes of the chamber, we placed industrial 3-Megapixel cameras (Basler acA2040-120uc) with zoom lenses (opening angles: top view: 45° × 26°, back view: 55° × 26°, side view: 26° × 35°) and exposure times of 3 ms. To visualize nests in the dimly lit nest boxes, we used monochrome infrared cameras (2 MP, Basler daA1600-60um) and fisheye lenses (143° × 112°). The frame rate of the video is about 48 frames per second (frame period 21 ms). The spatial resolution of the main cameras is about 2.2 pixels/mm.

With the cameras, we filmed the arena directly from the top and indirectly from two sides via two tilted mirrors in front of the glass side panels. This setup yields a large object distance of about 1–1.5 m, which allows for a small perspective distortion. We relayed the uncompressed camera outputs (approx. 400 MB/s in total) to a host computer via a USB3 Vision interface. Each camera received frame trigger signals generated by the radio interface.

On the host computer that controlled the acquisition system, we ran two custom applications: BirdVideo, which acquires and writes the video data to a file, and BirdRadio, which acquires and writes the microphone and transmitter signals to a file (see Extension section, ‘Software and data’). The generated file pairs are synchronized such that for each video frame there are 512 audio samples (an audio frame) and 512 transmitter signal samples (a radio frame). While the recording is gapless, it is split into files of typically 7 mins duration.

BirdVideo is written in C++, makes use of the OpenCV library (Bradski, 2000) and FFMPEG (Tomar, 2006) and undistorts the nest camera images with a fisheye lens model and transforms the five camera images into a single 2,976 × 2,016 pixel-sized composite image (Fig. 1B). The composite images we compress with the h264 codec on a NVIDIA GPU and save the video as an MP4 (ISO/IEC 14496-14) file. We used constant-quality compression with a variable compression ratio in the range 150–370, resulting in a data rate of 0.72–1.8 MB/s, depending on the number of birds and their activity in the arena. Compression ratios in this range do not significantly decrease key point tracking performance (Mathis & Warren, 2018).

Microphones

We installed four microphones on the four side walls of the isolation chamber and one microphone on the ceiling. We mounted two more microphones in the nest boxes and one microphone outside the isolation chamber.

Antennas

We mounted four whip antennas (referred to as A,B,C, and D) of 30 cm length (Albrecht 6157) perpendicular to the metallic sidewalls and ceiling of the chamber (using magnetic feet). We fed the antenna signals to a USRP containing a large FPGA (Fig. 7A).

Figure 7 Schematics and operation of the radio receiver with its PLL demodulators and phase compensation.

(A) Schematic of the software-defined radio receiver. Analog antenna signals are first down converted by mixing the amplified signals with a local oscillator of frequency ωLO. The resulting intermediate signals have digital representations zA (t), zB (t), zC (t), and zD (t). From these, eight FM demodulators extract the transmitter tracking frequencies ωi (t), i ∈ {1,…,8}. (B) The power spectral density of the 80 MHz wide intermediate band zA (t) centered on the oscillator frequency ω“LO” = 300 MHz. The limiting ranges (gray bars) of the eight channels (transmitters) are typically set to ± 1 MHz of their manually set center frequencies (black vertical lines). A zoom-in (lower graph) to the limiting range of one channel reveals the tracking frequency ω(t) (light grey vertical line) that tracks the large peak at the transmitter frequency. Fs stands for full scale. (C) The baseband power spectral density uA (t) is a down-converted, ±100 kHz wide (dashed vertical lines), flat band around the tracking frequency (solid vertical line) close to the transmitter frequency (the large peak). (D) Schematic of one demodulator: a PLL (orange shading) computes the tracking frequency omega and four phase compensation circuits (blue shading) align the baseband signals ua (t) which are derived from intermediate signals with digital down-converters (DDCs) operating on a common tracking frequency omega. (E) Vector diagram in the complex plane illustrating the effects of the PLL (middle) and of phase compensation (right) on main and baseband vectors (shown only antenna A & B). The phase compensation circuits drive the phases αa = arg<ra>( ) of rotated baseband signals ra = ua e(−i*θ) to zero. (F) The four baseband vectors ua are shown without phase compensation (left), after aligning the main vector with the PLL (middle), and after additional phase compensation (right). The result is that all vectors are aligned, and the master vector is of maximal amplitude.

The bird arena

Inside the chamber, we placed the 90 cm × 55 cm (floor size) bird arena. To minimize acoustic resonances and optical reflections, we tilted the four 40-cm high side panels of the arena. We tilted opposite panels by 11° and 13° toward the inside, respectively. Two of the side panels are made of glass for videography and the two opposite panels are made of perforated plastic plates. We covered the floor of the arena with a sound-absorbing carpet. The upper part of the arena we enclosed with a pyramidal tent made of a fine plastic net that reached up to 125 cm above the arena. At a height of 35 cm, we attached two nest boxes to the side panels, each equipped with a microphone and a camera. On the floor, we placed perches, a sand-bath, and food and water trays.

Transmitter device

Our transmitter devices are based on the FM radio transmitter circuit described in Ter Maat et al. (2014) (Fig. 2). To distinctly record birds’ vocalizations irrespective of external sounds, we replaced the microphone with an accelerometer (Anisimov et al., 2014) that picks up body vibrations and acts as a contact microphone. The accelerometer (Knowles BU21771) senses acceleration with a sensitivity of 1.4 mV/(m/s2) in the 30 Hz–6 kHz frequency range (Knowles Electronics, 2017). Inspired by Ter Maat et al. (2014), we performed the frequency modulation of the accelerometer signal onto a radio carrier using a simple Hartley oscillator transmitter stage with only one transistor and a coil that functions both as the inductor in the resonator circuit and as the antenna. We set the carrier frequency ωc of the radio signal in the vicinity of 300 MHz, which corresponds to an electromagnetic wavelength of about 1 m.

The total weight of the transmitter including the harness and battery is 1.5 g, which is ca. 10% of the body weight of a zebra finch. The transmitters are powered by a zinc-air battery (type LR41), which lasted more than 12 days.

The radio circuit on the transmitter uses a single transistor and an inductor coil as emitting antenna to modulate the measured acceleration signal aT(t) (where T stands for Transmitter and t is the time) onto a (relatively stable) radio carrier frequency ωc(t). The accelerometer signal aT(t) is encoded as (instantaneous) transmitter frequency ωT(t)≃ωc(t)+caT(t) with c some constant. The high-pass filtered tracking frequency ωα(t) (with a cutoff of 200 Hz) is our estimate of the accelerometer signal aT(t) (Figs. 3, 4, 5 and 6). We refer to it as the demodulated (transmitter) signal.

Before mounting a device on a bird, we adjusted the coil to the desired carrier frequency in the range of 250–350 MHz by slightly bending the coil wires. Thereafter, we fixed the coil and electronics in dyed epoxy. The purpose of the dyed epoxy is to help identify the birds in the video images, protect the printed circuit board, and stabilize the coil’s inductance (Figs. 2B and 2C). We mounted the devices on birds using a rubber-string harness adapted from Alarcón-Nieto et al. (2018) (Fig. 2D).

Multi-antenna radio signal demodulation

Stage 1: Multi-antenna radio signal reception

We fed the antenna signals to a universal software radio interface (USRP-2945; National Instruments, Austin, TX, USA), which comprises four independent antenna amplifiers, the gains of which we set to 68 dB, adjustable in the range −20 dB to +90 dB.

Stage 2: Intermediate band

At the input stage of our demodulation technique, the radio interface generates from the amplified radio signal a down-converted signal. The radio receiver filters out an 80 MHz wide band around the local oscillator frequency ωLO, which we typically set to 300 MHz.

These four signals then become digitally available on the FPGA as complex valued signal of 100 MHz sampling rate. We call them the intermediate signals za(t),a∈{A,B,C,D} (intermediate band in the frequency domain; Fig. 7B). On the FPGA, they are 100 MS/s complex-valued signals of 2 × 18 bits precision. For details about the analog down conversion and the sampling of complex valued signals, see the National Instrument manual of the USRP and the documentation of our custom radio software.

Stage 3: digital down-conversion and tracking

On the FPGA, we instantiated eight demodulators that simultaneously demodulated up to eight transmitter channels. Each demodulator contains four digital down-converters that extract from the four intermediate signals za(t) the four baseband signals ua(t), one for each antenna a∈{A,B,C,D}. Because we accommodate up to eight transmitters, there are in total 32 baseband signals uia(t), one for each antenna and transmitter device.

The baseband signal for transmitter device i∈{1,...,8} is extracted around the tracking frequency ωi(t) (Figs. 7C and 7D). The decimation filter of this down-conversion has a flat frequency response within a ±100kHz band. Given the decimation factor of 128, the complex baseband signals uia(t) have a sample rate of 781.25 kHz and a precision of 2 × 25 bits. Since the digital signals are complex valued, we interchangeably call them vectors and their phases we refer to as angles.

The tracking frequency ωi we set with PLLs, one PLL for each transmitter (channel) i. A PLL generates an internal oscillatory signal of variable tracking frequency ω(t); it adjusts that frequency to maintain a zero phase with respect to the received (input) signal. As long as the zero-phase condition is fulfilled, the PLL’s tracking frequency ωi(t) follows the instantaneous transmitter frequency ωT,i(t) of the i′th transmitter. Each PLL generates its tracking frequency ωi by direct digital synthesis with a 48-bit phase accumulation register and a lookup table.

A PLL is driven by the main vector uiM(t)=uiA(t)+uiB(t)+uiC(t)+uiD(t) that we formed as the sum of the four baseband vectors. The error signal used in the PLL’s feedback controller is given by the phase of the main vector θi(t)=arg(uiM(t)) (Fig. 7D).

The tracking frequency ωi is dynamically adjusted to keep the phase of the main vector close to zero. We implemented the calculation of the angle θi=arg(uiM) of the main vector with the CORDIC algorithm (Volder, 1959) and unwrapped the phase up to ±128 turns before using it as the error signal for a proportional-integral-derivative (PID) controller that adjusts the tracking frequency of the PLL.

The PID controller was implemented on the FPGA in single precision floating point (32-bit) arithmetic. The controller included a limiting range and an anti-windup mechanism (Astrom & Rundqwist, 1989). The unwrapping of the error phase was crucial for the PLL to quickly lock-in and to keep the lock even during large and fast frequency deviations of the transmitter. To tune the PID parameters, we measured the closed-loop transfer function of the PLL by adding a white-noise signal to the control signal (tracking frequency), and then we adjusted the PID parameters until we observed a closed-loop transfer function with low-pass characteristic. We achieved a closed loop bandwidth of about 30 kHz.

Stage 4: Phase compensation

To avoid destructive interference in the summation of the baseband vectors, we compensated their phases αia(t)=∡(uiM(t),uia(t)),a∈{A,B,C,D} relative to the main vector. We introduced individual phase offsets (also called compensation phases) Δφia(t) that were set in feedback loops to drive the phases αia(t) towards zero (Fig. 7E).

For a given PLL i, we compensated the relative phases under which a radio signal arrives at the four antennas a∈{A,B,C,D} to align the four baseband vectors uia. The alignment was achieved by providing a phase offset Δφia to each down-converter, where it acts as offset to the phase accumulation register of the direct digital synthesis. To compute the angle of a baseband vector relative to the main vector, we rotated the baseband vectors uia by the phase θi of the main vector to result in the rotated vector ria=uiae−iθi. After averaging the rotated vectors across 512 samples, we computed its angle αia=arg(⟨ria⟩512). We then compensated that angle by iteratively adding a fraction γ∈[0,1] of it to the phase offset: Δφia:=Δφia+γαia. The parameter γ is the phase compensation gain (Fig. 7D), typically set to γ=0.2.

The PLL and phase compensation form independent control loops. When the PLL is unlocked (e.g., off), the tracking frequency does not match the instantaneous transmitter frequency and the baseband vectors rotate at the difference frequency (Fig. 7F, left). When the PLL is switched on and locked, the baseband vectors do not rotate, and the main signal displays a phase θi(t)≃0 (Fig. 7F, middle). When the phase alignment is switched on, the baseband signals align, and their sum maximizes the magnitude of the main vector (Fig. 7E, right).

Because birds’ locomotion is slower than their rapid vocalization-induced vibratory signals, the phases αia(t)change more slowly than the tracking frequency ωi(t) and therefore we updated phase offsets less often than the PLL’s tracking frequency at a rate of about 1.5 kHz (781.25 kHz/512).

Operation

The intermediate band of za(ω)is wide enough (80 MHz) to accommodate up to eight FM transmitters. Provided the transmitters’ FM carrier frequencies are roughly evenly spaced, the transmitter frequencies do not cross, even during very large frequency excursions. Nevertheless, we limited the tracking frequency ωi of channel i to the range [Ωi−ΔΩ,Ωi+ΔΩ], where Ωi is the center frequency and ΔΩ=1 MHz is the common limiting range of all channels (Fig. 7B). The center frequency Ωi of a channel we manually set at the beginning of an experiment to the associated FM carrier frequency. The limiting range of 1 MHz we found narrow enough for the PLL to rapidly relock after brief signal losses. However, when the PLL loses tracking, the tracking frequency ωi(t) always stuck at one of the two limits of its range. Therefore, we detected tracking loss events heuristically as the time points t that satisfied |ωi⁡(t)−Ωi|>=Δ⁢Ω.

Software and data

Synchronization

To ensure that the data streams from the video cameras, microphones, and transmitter devices are synchronized, we recorded videos with industrial cameras and digitized sounds on a National Instruments data acquisition board, both of which received sample trigger signals from the USRP. The base clock of the USRP was 200 MHz and was divided by 213 to generate the audio sample rate of 24.414 kHz, and with a further division by 29, we generated the 47.684 Hz video frame rate.

Central control software

On the host computer we run our central control software (BirdRadio programmed with LabView) that acquires the microphone and transmitter signals and writes them to a TDMS file (Fig. 1C). BirdRadio also logs both power and variance trajectories of the baseband signals ua(t) and ra(t) associated witch a given transmitter, which allows us to evaluate the radio signal-to-noise ratio RSNRa and RSNRM of both single-antenna signals ( a) and multi-antenna ( M) signals on the same dataset (see Section Radio Signal-to-Noise Ratio).

Furthermore, BirdRadio sends user datagram protocol (UDP) control signals to BirdVideo, it automatically starts the recording in the morning and stops it in the evening, it controls the light dimming in the sound-isolation chamber with simulation of sunrise and sunset, it triggers an email alarm when the radio signal from a transmitter device is lost, and it automatically adjusts the center frequency of each radio channel every morning to adjust for carrier frequency drift.

Data management

The BirdPark is designed for continuous recordings over multiple months, producing data at a rate of 60 GB/day for two birds and 130 GB/day for eight birds. On a subset of the data analyzed here we implemented the FAIR (findable, accessible, interoperable, and reusable) principles of scientific data management (Wilkinson et al., 2016) as follows:

Our recording software splits the data into gapless files of 20,000 video frames (ca. 7 min duration). At the end of a recording day, all files are processed by a data compilation script that converts the TDMS files into HDF5 files and augments them with rich metadata. The HDF5 files are self-descriptive in that they contain metadata as attributes and additionally, every dataset in the file contains its description as an attribute. We use the lossless compression feature of HDF5 to obtain a compression ratio of typically 2.5 for the audio and accelerometer data. The script also adds two microphone channels to the video files. Although this step introduces redundancy, the availability of sound in the video files is very useful during manual annotation of the videos. Furthermore, the script also exports the metadata as a JSON file and copies the processed data onto a network attached storage (NAS) server. At the end of an experiment, the metadata is uploaded onto our internal openBIS (Bauch et al., 2011) server and is linked with the datafiles on the NAS.

Technical measurements

Sound attenuation

The isolation chamber attenuates sounds by 30 dB, which we measured by playing a 3-s continuous white noise stimulus on a loudspeaker inside the chamber and by recording it both inside and outside the chamber. We defined the sound attenuation as the difference in log-RMS amplitude in the 400 Hz to 5 kHz frequency band (corresponding roughly to the vocal range of zebra finches).

The background acoustic noise level inside the chamber is roughly 37 dBA (measured with a Voltcraft SL-10 sound-level meter). The acoustic reverberation time of the chamber inside is 36 ms: First we measured the impulse response function by playing a white noise signal on a loudspeaker and recording the sound with a microphone, both inside the chamber and then correlating the two signals. We estimated the reverberation time as the exponential decay time of the squared impulse response function over a decay range of 60 dB.

Transmitter properties

We found that the carrier frequency ωc of FM transmitters depends on temperature at an average of −73 kHz/°C (range −67 to −87 kHz/°C, n = 3 transmitters). Towards the end of the battery life (over the course of the last 3 days), we observed an increase of ωc by an average of 500–800 kHz. These slow drifts can easily be accounted for by tracking and high-pass filtering the momentary frequency. The measured end-to-end sensitivity of the frequency modulation is 5 kHz/g, with g being the gravitational acceleration constant.

Radio signal-to-noise ratio (RSNR)

While animal carried an operational transmitter, we calculated the radio signal-to-noise ratio RSNRa(tk) of the signal ra(t) with a∈{A,B,C,D,M} in the (non-overlapping) k-th 21-ms radio frame Fk defined by its center time tk as a function of the signal power (at zero frequency): pa(tk)=|⟨ra(t)⟩(t∈Fk)|2 (where averages ⟨⋅⟩ are calculated over all samples in a radio frame and the noise variance va(tk)=⟨|ra(t)|2⟩(t∈Fk)−|⟨ra(t)⟩(t∈Fk)|2:

RSNRa(tk)=10log10[pa(tk)va].

To obtain smooth estimates of the variance va(tk), we estimated the latter as the median value va across all n = 20,000 radio frames per 7-min long file. In Fig. 3A, we compared the multi-antenna RSNRM(tk) to the RSNR∗(tk) of the best single antenna defined as:

RSNR∗(tk)=maxa∈{A,B,C,D}⁡⟨RSNRa(tk)⟩k.

In Fig. 3A, the RSNRM measurements are roughly constant (max. deviation of RSNR of 1 dB) in each of the examples shown on top. We estimate the noise power PN of the (high-pass filtered) demodulated signal as the power spectral density (PSD) averaged over periods without vocalization and integrated from 0 to 8 kHz (Fig. 3A). In Fig. 3B, we plot PN against RSNRM on the noise segments in Fig. 3A and on additional n = 216 noise segments within periods in which RSNRM was roughly constant; these noise segments were selected as follows from a large dataset. First, we selected segments in which RSNRM was inside intervals [kdB,(k+1)dB],k∈[1,…,50] for at least 100 ms. Second, segments that contained apparent signal components generated by the bird were removed by manual inspection of the spectrogram.

In Fig. 3C, we compare signal-to-noise ratios of multi vs single-antenna demodulation. In theory, when the PLL locks (when phases are aligned and ra(t) equals ua(t)) and assuming independence of radio amplifier noises, the noise power (variance) of the phase-compensated and summed signal uM(t) is four times larger than that of a baseband signal ua(t) for a given antenna and the signal power of uM(t) is 16 times larger than that of a single antenna signal ua(t). Taken together, we expect multi-antenna demodulation to increase the RSNR by 6 dB.

Detection of radio frequency jumps

We detected jumps of the transmitter tracking frequency ωi as frequency deviations larger than 50 kHz from the running median, computed within 0.2 s sliding windows. Frequency jumps were concatenated when they were separated by less than 50 ms. The modulation (or jump) amplitude was defined as the maximum absolute deviation.

Detection of wing flaps and video alignment

The transmitter signals evoked by proximity effects during wing flap movement were downward and dip-like (Fig. 6). We detected dips on the male’s transmitter signal by thresholding its derivative after down-sampling by a factor of 32. A wing flap event was recognized when the derivative initially falls below a negative threshold, −W = −60 MHz/s, and subsequently rose above a positive threshold, W within a 2–12 ms window following the initial threshold crossing. The detected dips corresponded closely with the center time point of the wing’s down stroke (Fig. 6B).

We time-stamped the video frames as follows. For each detected wing-flap event, we examined the transmitter’s signal range. Events associated with a signal range less than 200 kHz we discarded from further analysis, where we defined the signal range as the maximum minus the minimum transmitter signal in a time window [−10.5, 10.5] ms (corresponding to 1 video frame period) centered on the minimum of the signal dip. Similarly, we excluded abrupt signal changes exceeding 1,000 kHz.

The video frame with exposure onset nearest to the dip minimum we defined as the first event frame, and the subsequent frame as the second event frame. We then ordered all event frames according to their time stamp: the time lag between the exposure onset and the dip minimum.

By randomly sampling video frames in which the flapping birds were clearly visible and continuously flying (from n = 5 birds), we obtained the illustration of wing flapping shown in Fig. 6, nicely capturing the phases of the wing flap cycle. All the frames in Fig. 6 are derived from a comprehensive dataset of five birds, with each event capturing a wing flap from a randomly selected bird.

Vocal analysis

Manual voice activity detection

In each of two bird-pair experiments (copExpBP08 and copExpBP09), we manually segmented all vocalizations of both birds in a randomly chosen file of 7 min duration. In experiments with four birds (juvExpBP01) and eight birds (juvExpBP03), we segmented all bird vocalizations in the first 1 min each of seven randomly chosen files (to obtain more diversity). We excluded files with a signal loss rate above 0.1%. We high-pass filtered (with a passband frequency of 200 Hz) the raw demodulated transmitter signals and microphone signals and produced spectrograms (in windows of 384 samples and hop size of 96 samples) that we manually segmented using Raven Pro 1.662 or BpBrowser (our custom data annotation software). We performed three types of vocal segmentations in the following order: (1) Transmitter-based vocal segments: On all transmitter channels, we separately segmented all vocalizations, precisely annotating their onsets and offsets. In the released data sets, these segments are referred to as transmitter-based vocal segments. We visually checked for crosstalk (whether we saw a faint trace of a vocalization on another bird’s transmitter spectrogram). We excluded crosstalk segments from further analysis (except for the crosstalk statistics): If a segment was detected as crosstalk, we set the label Crosstalk to the transmitter channel of the bird that generated the crosstalk, otherwise we set the tag to ‘No’. When we were uncertain whether a sound segment was a vocalization or not, we also looked at spectrograms of microphone channels and listened to sound playbacks. If we remained uncertain, we tagged the segment with the label ‘Unsure’: we treated such segments as (non-vocal) noises and excluded them from further analysis (except for the uncertainty statistics).

(2) Microphone-based vocal segments: We simultaneously visualized all microphone spectrograms (we ignored nest mics when the nest was not accessible to the birds) using the multi-channel view in Raven Pro (we ignored Mic7, which was located in the second nest that was not accessible to the birds). On those, we annotated each vocal segment on the first microphone channel on which it was visible (e.g., a syllable that is visible on all microphone channels is only annotated on Mic1). Overlapping vocalizations were annotated as a single vocal segment. When we were uncertain whether a sound segment was a vocalization or not, we also looked at spectrograms of transmitter channels and listened to sound playbacks. If we remained uncertain, the segment was tagged with the label ‘Unsure’, such segments were treated as (non-vocal) noises and were excluded from further analysis.

(3) Consolidated vocal segments: All consistent (perfectly overlapping down to a temporal resolution of one spectrogram bin) transmitter- and microphone-based vocal segments, we labelled as consolidated vocal segments. We then inspected all inconsistent (not perfectly overlapping) segments by visualizing all channel spectrograms. We fixed inconsistencies that were caused by human annotation errors (e.g., lack of attention) by fixing the erroneous or missing transmitter- and microphone-based segments. From the inconsistent (partially overlapping) segments that were not caused by human error, we generated one or several consolidated segments by trusting the modality that more clearly revealed the presence of a vocalization (hence our reporting of ‘misses’ in Table 2).

In our released annotation table, we give each consolidated vocal segment a Bird Tag (e.g., either ‘b15p5_m’ or ‘b14p4_f’) that identifies the bird that produced the vocalization, a Transmitter Tag that identifies the transmitter channel on which the vocalization was identified (either ‘b15p5_m’ or ‘b14p4_f’ or ‘None’), and a FirstMic Tag that identifies the first microphone channel on which the segment was visible (‘Mic1’ to ‘Mic6’, or ‘None’). We resolved inconsistencies and chose these tags as follows: If a microphone (-based vocal) segment was paired (partially overlapping) with exactly one transmitter segment, a consolidated segment was generated with the onset time set to the minimum onset time and the offset time set to the maximum offset time of the segment pair. The Bird and Transmitter Tags were set to the transmitter channel name, and the FirstMic Tag was set to the microphone channel name.

If a transmitter segment was unpaired, a consolidated segment was created with the same onset and offset times. The Bird and TrCh Tags were set to the transmitter channel name, and the FirstMic Tag was set to ‘None’.

If a microphone segment was unpaired, we tried to guess a transmitter channel name based on the vocal repertoire and noise levels on both transmitter channels. We visually verified that the vocal segment was not the result of multiple overlapping vocalizations (which was never the case). Then we created a consolidated segment with the same on- and offset, set the FirstMic Tag to the microphone number, the Bird Tag to the guessed transmitter channel name or ‘None’ if we were unable to make a guess, and the Transmitter Tag to ‘None’.

If a microphone segment was paired with more than one transmitter segment, a consolidated segment was created for each of the transmitter segments. The onsets and offsets were manually set based on visual inspection of all spectrograms. Bird and Transmitter Tags were set to the transmitter channel name and the FirstMic Tag was set to the first microphone channel on which the respective segment was visible.

We never encountered the case where a transmitter segment was paired with multiple microphone segments.

Vocal statistics

The statistics in Table 2 were calculated as follows:

# vocalizations = (number of consolidated segments)

# voc. missed on tr. channel = (number of consolidated segments with Transmitter=None)

# voc. missed on all mic. channels = (number of consolidated segments with FirstMic=None)

# voc. missed on Mic1 = (number of consolidated segments with FirstMic≠Mic1 and FirstMic≠None)

# voc. unassigned = (number of consolidated segments with Bird=None)

# voc. with overlaps = (number of consolidated segments which overlap with at least one other syllable by at least one spectrogram bin)

# voc. with crosstalk = (number of transmitter-based segments with Crosstalk)

# uncertain segments = (number of transmitter-based and microphone-based segments with ‘Unsure’ tag)

Supplemental Information

Supplemental Information 1 Birdpark Software Visualization.

Supplemental Information 2 ARRIVE Checklist.

We thank Aymeric Nager for his help with the design and construction of the BirdPark and Moritz Wohlhauser for creating the illustration in Fig. 2D.

Glossary of terms

DDC with digital down-converters

FPGA Field-Programmable Gate Array

PID Proportional-Integral-Derivative (controller)

PLL Phase-Locked Loop

RSNR Radio Signal-to-Noise Ratio

USRP Universal Software Radio Peripheral

LED light emitting diodes

PSD power spectral density

Fs full scale

MXI Multisystem Extension Interface

USB3 Universal Serial Bus 3.0

PCIe Peripheral Component Interconnect Express

TDMS Technical Data Management Streaming

MP4 MPEG-4 Part 14

UDP User Datagram Protocol

Additional Information and Declarations

Competing Interests

The authors declare that they have no competing interests.

Author Contributions

Linus Rüttimann conceived and designed the experiments, performed the experiments, analyzed the data, prepared figures and/or tables, authored or reviewed drafts of the article, and approved the final draft.

Yuhang Wang analyzed the data, prepared figures and/or tables, authored or reviewed drafts of the article, and approved the final draft.

Jörg Rychen conceived and designed the experiments, prepared figures and/or tables, authored or reviewed drafts of the article, and approved the final draft.

Tomas Tomka analyzed the data, authored or reviewed drafts of the article, and approved the final draft.

Heiko Hörster performed the experiments, prepared figures and/or tables, improved animal welfare procedures, and approved the final draft.

Richard H. R. Hahnloser conceived and designed the experiments, prepared figures and/or tables, authored or reviewed drafts of the article, and approved the final draft.

Animal Ethics

The following information was supplied relating to ethical approvals (i.e., approving body and any reference numbers):

All experimental procedures were approved by the Cantonal Veterinary Office of the Canton of Zurich, Switzerland.

Data Availability

The following information was supplied regarding data availability:

Raw data and annotations are available at Zenodo: Linus Rüttimann, Jörg Rychen, Tomas Tomka, Heiko Hörster, Mariana Rocha, & Richard Hahnloser. (2022). Behavioral recordings of mixed-sex zebra finch pairs with vocal segment annotations (1.0.0) [Data set]. Zenodo. https://doi.org/10.5281/zenodo.7105196

TheMP4 videos and HDF5 files are available at Zenodo: Linus Rüttimann, Wang, Y., Jörg Rychen, Tomas Tomka, Heiko Hörster, Mariana Rocha, & Richard Hahnloser. (2024). Recordings of zebra finch group behaviors with manually annotated vocal segments [Data set]. Zenodo. https://doi.org/10.5281/zenodo.13144875

The source code is available at GitLab: https://gitlab.switch.ch/hahnloser-songbird/birdpark

The BirdRadio is available at the ETH Zurich Library: http://doi.org/10.5905/ethz-1007-764

The BirdVideo is available at the ETH Zurich Library: http://doi.org/10.5905/ethz-1007-763.

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
