# Peer review of "Multimodal system for recording individual-level behaviors in songbird groups"

_PeerJ, doi:10.7717/peerj.20203_

## Round 0.1 · original submission · Major Revisions

· Academic Editor

Major Revisions

Thank you very much for your manuscript titled “Multimodal system for recording individual-level behaviors in songbird groups” that you sent to PeerJ.

This study presents an innovative technical-methodological concept for the study of bird behavior.

As you will see below, comments from reviewers 1 and 2 suggest a minor revision while reviewer 3 suggests a major revision before your paper can be published. Given this, I would like to see a mayor revision dealing with the comments. Their comments should provide a clear idea for you to review, hopefully improving the clarity and rigor of the presentation of your work. I will be happy to accept your article pending further revisions, detailed by the referees.

Reviewer 1 has some technical comments related to the equipment used and technical concepts, as well as the behavior of birds within the experimental design.

Reviewer 2 has suggestions for improving the written presentation of the work related to writing and presentation style, as well as the definition of some technical concepts.

Reviewer 3 has several comments for each section of the manuscript that are intended to improve the overall presentation and flow of the entire manuscript, especially given the amount of technical details that need to be included.

Please note that we consider these revisions to be important and your revised manuscript will likely need to be revised again.

·

Basic reporting

Songbirds are excellent animal models for neuroscience studies of sensory-motor integration, learning, and skilled motor sequence production. To obtain behavioral and neurophysiological data, researchers commonly house individual birds in isolation. But, this practice does not allow researchers to observe important processes such as song tutoring and courtship – behaviors in which song is used in a social context and require longitudinal monitoring. Prologued housing in isolation also raises ethics concerns that do not arise in group housing. Rüttimann et al. present a novel multimodal system for tracking individual behaviors in songbird groups using a combination of audio, video, and radio-based transmission of accelerometry recordings of individual birds. The key strengths of the work are 1) a system that successfully combines video, audio, and sensor data into a synchronized dataset, one of the main limitations introduced, 2) evaluation of a multi-antenna system for sensor data transmission, and 3) very low ambiguity about animal ID in the multi-modal data.

The study presents an innovative methodological framework and is within the journal's scope. The research question is well defined, and the multimodal approach is a significant contribution to behavioral tracking in birds.

The writing is of high quality, and the background literature is well-referenced. Figures are high quality and relevant. A minor revision of the text can improve readability for a broad audience: The manuscript is highly technical, particularly in its radio communication engineering aspects. Given the interdisciplinary nature of the study, text revisions should be made to explain concepts such as phase compensation, multi-antenna demodulation, and the rationale for using low-frequency transmission. Providing additional background or references for readers unfamiliar with software-defined radio techniques would improve accessibility.

The following sections detail key issues and some relatively minor experiment and analysis concerns that can be addressed in a minor revision of the manuscript.

Experimental design

1. Justification for Low-Frequency Transmission: The choice of the radio frequency should be better justified. The authors mention considerations of weight, battery life, size, and they mention issues od RF signal fading. All of these also relate to the choice of transmitter carrier frequency. The current explanation is insufficient.

2. Bird Behavior in the Setup: The manuscript lacks data on the singing behavior of the birds within the experimental setup. Did the harnesses or transmitters affect singing frequency, motivation, or social interactions? Addressing this is important for evaluating ecological validity. The authors reference another study with a different setup where behavior was quantified. It should be straightforward to do the same here.

Validity of the findings

1. Ground Truth Validation: The validation of the method for identifying vocalizations would be stronger if the authors provided a ground truth benchmark. One possible approach is using loudspeakers to play known vocal sequences, controlling the volume, and confirming whether the system correctly attributes these to the simulated "birds." Without such validation, it is difficult to assess the absolute accuracy of the segmentation and the rate of missing vocalizations cannot be trusted due to this experimental design issue. The authors recognize this limitation in their manuscript but do not address it.

2. Multi-modal integration: A key advantage of multi-modal system is the integration of audio and video data for tracking motion and identifying sound sources. The authors discuss the use of microphone arrays for source separation. The analysis in the manuscript evaluates cross-talk between microphones and accelerometers but does not evaluate multi-modal integration of vocalizations. Instead, this integration is demonstrates with capturing wing flapping. The wing flapping analysis is an extreme example of multimodal synchronization but it diverges from the required strengths of the system. A more behaviorally relevant test case would be vocal interactions, where video and accelerometry could provide valuable insights into turn-taking and coordinated movement of multiple birds.

Additional comments

1. Line 148: It’s better to report the background noise in dBSPL (or dBPa). dBA includes weighting of frequency bands by the human loudness perception and that is irrelevant for birds. Were fans turned off in this measurement? 37dBA is low.
2. Line 119: Consider clarifying ‘clock dividers’ to help general readers.
3. Line 150: The method of measuring reverberation time is not clear. Was an impulse played? Or a constant white noise that was stopped?
4. Line 235: The abbreviaion PLL is used before it is defined. A brief glossary or expanded explanations within the text would help readers unfamiliar with these terms.
5. Section on Radio signal-to-noise ratio: Please add text to clarify the calculation. It is not self-evident that this is how the SNR is calculated.
6. Lines 517-528: Could there be inductive crosstalk between transmitters? In close proximity between birds, is it possible that one transmitter is received in another?
7. Line 391: It is not clear if video frames were interpolated to get resolution that is better than the frame rate.
8. Figure 6a: Red and purple are similar colors for marking male and female vocalizations. Choosing more distinct colors is better.

·

Basic reporting

Most of the manuscript is intelligible, however in several places you have jargon that will be unfamiliar to even a fairly specialized audience without properly defining it. For example, PLL and USRP are both used before being defined. LC Resonator is brought up without being described ever. I think this is a real problem for your work, since your manuscript would likely to be of great interest to songbird researchers, most of whom don’t have a strong background in the details of radio frequency transmitters.

I also think the introduction could provide a bit more background information so that people (myself included) will have a better idea of the state of the field for radio transmitters as well as automated tracking of vocal/non-vocal behavior, so that people can appreciate, for example, your custom handing of multiple-antenna for radio frequency. Basically I think for every main product of this manuscript (which I see as the software itself, the specific RF methods, and the clever integration of multimodal data-streams), you could provide just a bit more background to explain whether/why these things are useful and what the necessary alternative would be.

Additional, and this is partly style preference, but I think you use far too much passive voice throughout. I think the paper would be much more readable if you work to favor active voice and more direct verbs over the complex, indirect phrasing you have currently.

There’s also some redundancy in your methods and results, so I think you could check to make sure you’re not repeating yourself.

Experimental design

I addressed this above, but as it is, it’s somewhat unclear what the take-away of this manuscript should be. Is this just a methods paper that you yourself will be able to cite when you do the behavioral experiments, or do you expect other people to take something away from this? Do you think other people should use BirdPark? Should they handle multi-antenna RF tracking in similar ways that you do? Should they accelerometers plus cameras to up-sample behavioral video? I think your demonstration of catching the wing-cycle, a behavior that is faster than your camera can capture, is very cool, but I think you could highlight why this is beneficial. Obviously wing flapping is easy enough to capture with a high speed camera, but there are plenty of behaviors that this would be really useful for. I think you should set that up in the intro and highlight it in the discussion.

In general, I think you provide excellent detail on your recording set up and your methods, and the code is available to look through. I didn’t get a chance to play around with it myself. Given that it’s not super straightforward to install and run BirdPark significant time investment, I think it would be helpful to provide some videos of what using the software itself looks like, so that people could consider whether that would be a good approach for their specific questions.

Validity of the findings

It’s hard to judge this without replicating this system, but as best as I can tell this is a really powerful system for tracking animal behavior, and is something that could be duplicated in another lab based on the information provided. Beyond this you provide useful approaches to dealing with common problems, so even if people don’t install and implement BirdPark specifically, they will be able to take advantage of your innovations here.

Additional comments

There are some places where formatting could be better. The spacing of the equations seems off. Unless this is the standard in the field, I really don’t love using * as a variable. The description of stats from the supplementary table also seems a little bit less professional, style-wise, than the paper as a whole.

Finally, as it stands, this paper feels like it’s being pulled in two directions: between being written generally for animal behaviorists and written very specifically for experts in radio frequency tracking. I think you should reflect a bit on who your intended audience is here. Obviously, I’m a biased by my own interests, but I personally think you should aim for a general, animal behavior audience. As an animal behaviorist, I found the RF sections very dense and hard to follow, which I think does a disservice to this work. I don’t think you need to remove all this detail, but I suspect you could move some of the technical details to a supplement. Whether or not you do this, I think you should provide a little bit more broad explanations so that those of us who might never need to understand the specifics of parsing multiple RF signals across multiple antennae will still be able to appreciate the problem you were trying to solve and the general theory behind what you’re doing.

All that said, this seems like really excellent work, all of these suggestions are really quite minor notes about how to improve the presentation, I don’t have any suggestions on the methods themselves. I’m really excited to see what you plan to do with it, and I think with some revisions of the manuscript, you could make this an extremely useful resource for people interested in tracking.


Also, apologies that this is not a more detailed review. I think this work is very good and I think the manuscript deserves careful revision. I’ve laid out some broad ideas, I wish I had more time to do more line-by-line notes, but I think it will be easy enough to apply suggestions throughout.

Reviewer 3 ·

Basic reporting

The authors present a technological tool ('BirdPark') to improve longitudinal studies of individuals for questions related to behavior and social communications in species. This manuscript has a lot of details that will provide readers with information on each component necessary to build such a system. On the other hand, what is missing is a clear outcome of how BirdPark worked for the tested groups. It is understandable that focusing on the questions for each bird group (for breeding or song studies) will be conducted in a separate study. However, not showcasing more detailed summaries for these groups obscures the outcome regarding how effectively BirdPark studies different group sizes. Specific questions that can help with improvement: how much data was pooled for each test? did it vary if the chamber was used to study pairs or groups of birds? what were the ranges of the video/transmitter/radio data for each group? did the multimodal framework perform better for a given group? Additionally, the longitudinal aspect is diminished throughout the methods and results sections. While it is part of the title and proposed as an objective in the introduction, it is not mentioned in detail until the discussion/conclusion. The methods and results require content that connects how the technologies specifically apply to longitudinal studies.

The general reporting and flow of the manuscript needs improvement, especially because of the level of technical details that need to be included which can result in losing track of the purpose of a section. I provide general comments to improve the general reporting in this manuscript:

The introduction covered the technological challenges of methods of behavior monitoring as well as their general advantages, leading up to the proposal of a multimodal method, 'BirdPark.' The introduction needs flow improvements so the reader can extract the main aspects of the three technologies (video, sound, sensor) more easily. For example, linking the third paragraph (lines 45-50) with the second (41-44) would create a larger paragraph encompassing the introductory information for video. Check if other paragraphs could be combined or linked or if there are topic sentences at the beginning of the paragraphs that could be improved for the flow of the introduction.

The methods sections could benefit from adding a level (sub-headings) to organize the components of the complete multimodal approach. Currently, each section describes a component, but it is up to the reader to identify which components correspond to the data collection, which to data processing, and which to output interpretation. Adding these levels would improve the understanding: Methods->Data collection or Multimodal hardware (Chamber, Video, Transmitter...) -> Data processing/management, etc.

The methods' subsections that describe the technical aspects of the multimodal approach are useful. To enhance the reader's understanding, especially for those who may find these methods highly technical due to a lack of experience with the equipment and its operation, I recommend beginning each subsection in a manner similar to the "Video Acquisition System" and "Transmitter Device" sections, where the goal is stated in the first sentence. For example, phrases like "To visualize the arena..." and "To sense acceleration" should be used. Where possible, briefly indicate what each component measures again. While this was introduced in the introduction, including a brief statement about what the transmitter device measures will assist researchers in understanding the purpose of each component.

The results combine the methods with the outcome, partly because the manuscript aims to propose a method. However, this section should emphasize the results. For instance, in the first paragraph, instead of starting with: "We built an arena optimized for..." change it to: "The arena we built optimized..." This opening statement should be followed by metrics or qualitative data that supports the results (minimized acoustic resonances, optimal audiovisual recording).

For the discussion section, the comparison between this method and the available or current frameworks used must be presented. If there is no similar technology to compare, because this is the first attempt to bring together these technologies into one framework that must be stated after conducting a fresh search of the literature. Additionally, even if other institutions or researchers have partially combined technologies, those can also serve as a comparison to what BirdPark does. Currently, the discussion begins indicating a result- the miss rate reduction- which is not an overarching result but a specific one. I recommend that the authors evaluate the outline of what they wish to discuss, starting with the key result(s), comparing them to existing methods and techniques, and concluding with remaining challenges, future improvements, steps, or implications of this method for the field of behavioral studies. Importantly, they should bring the discussion back to how this advancement will enhance the longitudinal aspect of these studies. Currently, the results and discussion section lacks sufficient detail to connect the two: multimodal advancements and longitudinal studies, especially for a broad readership of this journal.

The conclusions section includes content that is more appropriate for the discussion as it ties this work to potential applications. The conclusions section may be reduced to one paragraph within the end of the discussion or following it that indicates- what BirdPark is, what it does and why it's an alternative.

Each figure must include a figure caption. Currently there are only two captions for figures 3 and 5.

Experimental design

no comment

Validity of the findings

I mentioned in the general reporting section how the results section is currently methods heavy and it needs to be restructured to highlight the results using the data obtained from testing the BirdPark.

For a manuscript like this, which proposes an improved method, the improvements should be measurable and then compared (in the Discussion) to what is currently available. In other words the results section should make it clear how BirdPark performs to collect all of this data and improve future studies. Since BirdPark was tested on different groups of birds, a table or summary or brief statement of how the methods performed across groups is relevant to understand the range of applications of the method. Finally, review each section of the results to determine whether there is a measurable, reportable outcome or if the description provided in the methods suffices. For instance, the "Transmitter device" section references other works, which is more appropriate for a methods section. The writing in the "Radio receiver" section describes how the accelerometer signal was reconstructed instead of explaining what that accomplished or how it enhanced the data collection. If there is no clear result for each component, but rather an improvement in data collection when they are combined, then prepare a results section that guides us through how the components of the transmitter, radio, video, etc., previously outlined in the methods, come together to provide a clearer way of studying communication or behavior.

Authors must indicate for each component of BirdPark what the data outputs are at the beginning of the results or within each section. How many hours of video (the equivalent for the other components) per tested group were recorded.

Additional comments

Minor comments
- Within the paragraph of lines 125-131 introduce that the chamber has glass panels or specify the materials.
- Join paragraphs with lines 132-135 and 136-138. These two paragraphs are referring to the setup of the top part of the chamber (its components and what they are meant for). In general very short paragraphs (1-2 sentences) do not develop a full idea and break the information in a way that made it harder for me to follow the purpose of each paragraph.
- In lines 140-141 the nest boxes are mentioned for the first time. Mention the context of the chamber in previous paragraphs; it could be the first paragraph of the section "The Chamber". Indicate the non-technological components of the chamber and their objective. For example: Within the chamber you recreate an environment with nest boxes, etc. two study breeding behavior, etc.
- Lines 143-144. Change "roughly" for "on average", and provide averages and dispersion values (standard deviation, standard error or a range). These data are informative on the range of the conditions for the behavior experiments.
- Lines 145-147. Modify "roughly" in this sentence and throughout the manuscript (see line 149, 150, etc), instead indicate "on average", "the mean was", etc and present an average with dispersion values.
- Line 152. Modify to "We defined the reverberation time as..." Also consider rephrasing this sentence for a better interpretation of the measurement. Perhaps indicating, how to gauge reverberation time first, the smoothed squared impulse response was measured, followed by the next components of the metric.
- Line 156. Indicate the total number of panels. Then continue to describe their position, within the arena, angles,etc.
- Line 157. Specify if one panel is tilted at 11 degrees and the other at 13 degrees.
- Line 164. Modify the scientific name of the zebra finch to be in italics and lowercase the species part of the binomial name.
- Line 168. Re-organize the figure names. In this line the first figure referenced is Figure 2, but Figure 1 has not been referenced yet. Order the figures and their components (i.e. Figure1A-D, etc) so they appear in sequence.
- Line 170. Indicate after the amount of days of recording per group, if the groups were studied immediately one after the other or if there were resting periods within the study groups.
-Line 180. Cite or reference a source for the ARRIVE guidelines. Reference it supplemental file if you are going to include it.
- Lines 189-190. Is the "C" after the citation meant to be referencing a figure?
- Lines 199-200. These lines seem to be general descriptors of the video which may be best suited for the first paragraph of the "Video acquisition system"
- Line 203. Note the dash style used here and that used in other parts of the manuscript (line 196). Make changes to keep them consistent.
- Line 217-219. While the goal of this manuscript is not to provide insight on what was recorded and analyzed, it is relevant to note if the transmitter fits within the acceptable range of the weight for birds. Ten percent of the weight is higher than what has been proposed as adequate to have negligible effects on birds. (See: https://besjournals.onlinelibrary.wiley.com/doi/10.1111/j.2041-210X.2010.00013.x). Because this manuscript presents a multimodal approach which includes the use of transmitters, the justification or data to support that this approach does not cause negative effects that could hinder studies and inference needs to be presented here.
- Line 273. If BirdRadio is a widely available software, cite the source. Follow this prompt for all software cited.
- Lines 308-385. Section "Manual segmentation of vocalizations" is confusing because it does not specify if the manual segmentations were only done as a test for a subset of the individuals or if it was done for all individuals but only reported for the subset. Please indicate if/why this was only reported for a subset of birds. Additionally, the table in the supplemental files could be moved to the results section if these findings are indicative of how BirdPark performs or what still needs to be fine-tuned with the method. Show in the results section and discuss in the discussion section. Finally, edit the caption for the table so the first sentence indicates what the complete table is summarizing.
- Line 399. Change from "Events associated with a signal range less than 200 kHz we..." to "Events associated with a signal range less than 200 kHz were..."
-Lines 422-425 are not fitting for the results; these sentences describe what the transmitter does, which was covered/should be covered in the methods section.
- Lines 530-540. The start of this section "Performance of diversity combining technique" is more appropriate for a methods section of how the performance of the method was assesed. Recompose this section after assessing what content belongs in the methods and which one in the results.
- Include a caption for Table 1
- To enhance understanding of Figure 3, revise the first sentence to reflect its purpose: "Radio receiver with PLL demodulators and phase compensation to...." Conclude the prompt by stating the objective of this procedure. If readers consider this figure as standalone, it will provide all the information they need to comprehend it.

---

## Round 0.2 · accepted · Accept

· Academic Editor

Accept

After reviewing this revised version of your manuscript, I see that the main comments suggested by the reviewers have been included, while the suggestions not considered are justified in detail. Therefore, I am satisfied with the current version and consider it ready for publication.

·

Basic reporting

Overall, this is a strong revision that meets the publication criteria.

One very minor comment is the font size in some figures that is not legible.
(e.g. panels B,C,F in figure 7)

Experimental design

No comment.

Validity of the findings

No comment.

Additional comments

No comments.

·

Basic reporting

The revised manuscript is more clear and the presentation is improved generally.

Experimental design

The research question is more clearly defined and the approach is well suited to solve the exiting problems in studying songbird behavior and multimodal data streams generally.

Validity of the findings

This seems like a powerful approach and provides both a useful tool to implement, and a generally useful approach to emulate.